# Maternal Leucine-Rich Diet Minimises Muscle Mass Loss in Tumour-bearing Adult Rat Offspring by Improving the Balance of Muscle Protein Synthesis and Degradation

**DOI:** 10.3390/biom9060229

**Published:** 2019-06-13

**Authors:** Natália Angelo da Silva Miyaguti, Sarah Christine Pereira de Oliveira, Maria Cristina Cintra Gomes-Marcondes

**Affiliations:** Laboratory of Nutrition and Cancer, Department of Structural and Functional Biology, Biology Institute, University of Campinas, UNICAMP, Rua Monteiro Lobato, 255, Campinas, São Paulo 13083862, Brazil; namiyaguti@gmail.com (N.A.d.S.M.); sarah.pe9@gmail.com (S.C.P.d.O.)

**Keywords:** branched-chain amino acid, cancer cachexia, maternal influence, muscle protein turnover

## Abstract

Cachexia syndrome can affect cancer patients and new prevention strategies are required. Maternal nutritional supplementation can modify metabolic programming in the offspring, which lasts until adulthood. This could be a good approach against diseases such as cancer. A 3% leucine-rich diet treatment improved muscle protein turnover by modifying the mTOR and proteolytic pathways, thus we analysed whether maternal supplementation could ameliorate muscle protein turnover in adult offspring tumour-bearing rats. Pregnant Wistar rats received a control diet or 3% leucine-rich diet during pregnancy/lactation, and their weaned male offspring received a control diet until adulthood when they were distributed into following groups (n = 7–8 per group): C, Control; W, tumour-bearing; L, without tumour with a maternal leucine-rich diet; and WL, tumour-bearing with a maternal leucine-rich diet. Protein synthesis and degradation were assessed in the gastrocnemius muscle, focusing on the mTOR pathway, which was extensively altered in W group. However, the WL adult offspring showed no decrease in muscle weight, higher food intake, ameliorated muscle turnover, activated mTOR and p70S6K, and maintained muscle cathepsin H and calpain activities. Maternal leucine nutritional supplementation could be a positive strategy to improve muscle protein balance in cancer cachexia-induced muscle damage in adult offspring rats.

## 1. Introduction

Cachexia syndrome affects around 85% of advanced cancer patients, especially in some tumour types, such as stomach, pancreas, and colon cancer, being responsible for at least 22% of deaths, and still cannot be reversed [1]. Therefore, strategies that can minimise or even prevent this condition are valuable and necessary. Currently, nutritional supplementation as a preventive approach and the effects of maternal nutrition on the risk of their adult offspring risk developing a disease has been widely studied [2,3]. During gestation and lactation, maternal dietary composition can modify several aspects of offspring metabolic programming, which is considered to be the most important environmental factor affecting the expression of metabolic pathway-related genes in the offspring. Because these modifications can be preserved into adulthood, the maternal diet can be a preventive strategy against several diseases, such as cancer [4,5].

Cancer cachexia is a multifactorial syndrome that is characterised by progressive skeletal muscle mass wasting, with or without fat mass loss. Muscle wasting is related to a negative protein and energy balance because of the combination of reduced food intake and altered metabolism [6]. This abnormal metabolism involves an imbalance between muscle protein synthesis and degradation [7], impacting the mTOR pathway, which is important in the control of protein synthesis [8,9], and also interferes with the main degradation pathways in skeletal muscle, such as ubiquitin-proteasome, lysosomal, and calcium-dependent [10,11]. In an experimental model of cancer cachexia, a 3% leucine-rich diet showed positive effects on protein synthesis in skeletal muscle, leading activation of the mTOR pathway and reduction of protein degradation by inhibiting the ubiquitin-proteasome pathway in Walker-256 tumour-bearing rats [12,13,14]. Studies also showed attenuated degradation and modulation of both lysosomal and calcium-dependent proteolysis in animals fed this diet [15,16]. Considering the leucine supplementation capacity to cross de placental barrier and the ability to modify the breast milk composition [17,18,19], there is some evidence supporting the use of a maternal diet as a preventive approach. Our previous study showed that 3% leucine administered as a maternal supplement could modulate cancer cachexia-induced liver damages in tumour-bearing adult offspring [20].

Currently, no data have been reported on the preventive influence of maternal diet with leucine supplementation on cancer cachexia, especially in the skeletal muscle tissue. Therefore, the purpose of this study was to evaluate whether a maternal 3% leucine-rich diet might affect muscle wasting in adult offspring tumour-bearing rats. We hypothesised that this maternal diet can preserve muscle weight and improve the balance between muscle protein synthesis and degradation. This maternal supplementation modified these processes in the tumour-bearing rats. Therefore, we analysed the proteins related to the mTOR pathway, and mTOR and p70S6K activation were maintained. Regarding to the proteolysis pathways, the proteasome 20S subunit expression and the enzymes activities of cathepsin H and calpain were also preserved. Considering that the maternal environment influences the offspring stress response, we demonstrated that the maternal diet supplemented with leucine may be a positive strategy against cancer cachexia-induced deleterious effects in muscle mass of adult offspring.

## 2. Materials and Methods

### 2.1. Animals

Adult Wistar rats (male and female; 90 days old) were obtained from the animal facility at the Multidisciplinary Centre for Biological Research of the University of Campinas. The animals were kept in individually ventilated cages in the experimental room, which was located in the Laboratory of Nutrition and Cancer, under controlled environmental conditions (temperature, 22 ± 2 °C; light and dark, 12/12 h; and humidity, 50–60%). The rats were monitored daily, weighed three times/week, and received food and water ad libitum.

### 2.2. Diet

The diets were formulated in accordance with the American Institute of Nutrition (AIN-93; [21]). The control diet was the AIN-93 semi-purified diet, containing 18% protein. The leucine-rich diet was the same as the control diet except for the addition of 3% L-leucine. Therefore, the control diet contained 1.6% L-leucine, whereas the leucine-rich diet contained 4.6% L-leucine. The L-leucine dose was based on our previous experimental studies [13,14,15,16].

### 2.3. Experimental Procedure

The experiment was conducted according to the current ethical standards of the United Kingdom Coordinating Committee on Cancer Research [22] and was approved by the Ethics Committee on Animal Experimentation of the Institute of Biology at the University of Campinas (protocol number: #4224-1). The females were mated with the same male species (2 females:1 male) using the harem method [23], and after pregnancy was detected, the females were separated from the males. The pregnant rats were distributed into two groups (n = 3 per group) according to the diet administered, as follows: dams subjected to the control diet (DC) and dams subjected to the leucine-rich diet (DL). Throughout the gestation (21 days) and lactation (21 days) periods, the dams received the indicated diets, and the number of offspring was reduced to eight pups/litter.

To verify the influence of the maternal diet, after weaning, the male offspring (n = 7–8 per group) were fed the control diet. At 120 days of age, the adult offspring rats were distributed into four groups according to the presence of a tumour implant and the maternal diet provided, as follows: C, fed a control diet throughout the intrauterine, lactation, and adulthood periods, without a tumour; W, fed a control diet throughout the intrauterine, lactation, and adulthood periods, and tumour-bearing; L, fed a leucine-rich diet throughout the intrauterine and lactation periods, then fed a control diet until adulthood time, without a tumour; and WL, fed a leucine-rich diet throughout the intrauterine and lactation period, then fed a control diet until adulthood time, and tumour-bearing.

Food intake was measured, and the animals were weighed during the gestation and lactation periods (dams) and throughout their development (weaning until adulthood). The tumour-bearing rats (120 days old) received a tumour implant by subcutaneous injection of approximately 3 × 10^6^ viable Walker-256 tumour cells in the right flank [24,25]. After 21 days of tumour growth, all the animals were euthanised by decapitation. Blood was collected, and the gastrocnemius muscle and tumour were resected and weighed. The carcasses were then weighed. Figure 1 shows the protocol represented as a schematic timeline.

### 2.4. Cachexia Indicators: Serum and Morphometric Parameters

Serum parameters, including total serum proteins, albumin, cholesterol, and triglycerides, were quantified spectrophotometrically using commercial kits (Bioclin, Belo Horizonte, Brazil), according to the manufacturer’s instructions. The serum globulin was calculated by the difference between the total serum protein and albumin content. The carcass weight represents the whole-body weight without the liver, gastrocnemius muscle, and tumour.

### 2.5. Muscle Protein Degradation and Synthesis Analysis

Protein synthesis was assessed using aliquots of the gastrocnemius muscle that was immediately collected after the euthanasia, and the total L-[^3^H] phenylalanine uptake (Amersham Pharmácia Biotech do Brasil, São Paulo, Brazil) was measured, as previously described [13,16]. Protein synthesis rates were calculated by the amount of radioactivity incorporated into the gastrocnemius muscle and expressed as the count per minute (CPM) per hour per microgram of protein, normalised to 100 milligrams of muscle tissue. Protein degradation was measured in accordance with the methodology that was previously adapted in our laboratory [16,26], which consists of measuring of the amount of released tyrosine from muscle tissue after adding cycloheximide and a 2-h incubation. The protein degradation rate was expressed as the arbitrary fluorescence unit of released tyrosine content [27] per milligram of muscle protein per hour.

### 2.6. Enzymatic Assays

After euthanasia, the gastrocnemius muscle was removed, frozen in liquid nitrogen, and stored at −80 °C. Muscle samples were homogenised in phosphate buffer saline (PBS) and 0.1% Triton X-100 and centrifuged at 12,000 *g*. The supernatant was used to analyse protein content [28] and muscle enzymatic activity assays. The calpain activity was measured using casein as substrate and measuring at 595 nm the protein content with Coomassie brilliant Blue G-250 dye which not interact with the proteolytic products [29]. The chymotrypsin activity was measured by the release of aminomethly coumarin from the fluorogenic substrate N-Succinyl-Leu-Leu- Val-Tyr-7-Amido-4-Methylcoumarin (#S6510 succinyl-LLVY-AMC Sigma, Saint Louis, Missouri, USA) at 360 nm according to Orino et al. method [30]. The cathepsin H activity was measured using L-Arginine-7-amido-4-methylcoumarin hydrochloride (#A2027-Arg-NMec, Sigma, Saint Louis, Missouri, USA) as substrate in accordance with described by the Barret [31].

### 2.7. Western Blot Assay

The gastrocnemius muscle samples were homogenised in buffer containing 20 mM Tris, 1 mM DTT, 2 mM ATP, and 5 mM MgCl_2_, centrifuged at 12,000 *g*, and total protein content was assessed using the Lowry method [32]. Muscle samples (60 µg of protein) were analysed using SDS-polyacrylamide gel electrophoresis (10% or 12%) and transferred to a 0.45-µm pore size nitrocellulose membrane, which was blocked with skim milk (5%) for 1 h. Proteins were probed with primary antibodies GAPDH (SC47724) (Santa Cruz Biotechnology, Santa Cruz, CA, USA); 20S (PW8195), 19S (PW9265), and 11S (PW8185) (Enzo Life Sciences, Farmingdale, NY, USA); PI3K (4292), phosphor-PI3K (4228), mTOR (2972), phospho-mTOR (2971), p70S6K (9202), phospho-p70 S6KThr421/Ser424 (9204), 4E-BP1 (9452), phospho-4E-BP1Thr70 (9455), and eIF4G (2498) (Cell Signalling, Danvers, MA USA); and secondary antibodies goat anti-rabbit (7074) and horse anti-mouse (7076) (Cell Signalling). The Western blot band images were captured using the Alliance 2.7 (UVITEC, Cambridge, UK) and quantified using UVIband-1D (UVITEC), and protein expression was normalised using GAPDH as a loading control.

### 2.8. Statistical Analyses

For multiple group comparisons, the data were evaluated using a two-way ANOVA followed by the Bonferroni post-hoc test (Graph Pad Prism software, version 5.0, San Diego, CA, USA) [33]. For direct comparisons between two groups (especially between groups from different dams), the data were analysed using the Student’s *t*-test [33]. The results are expressed as the mean and standard error of the mean (SEM). Results were considered to be significant when the *p*-value was < 0.05.

## 3. Results

### 3.1. Maternal and Offspring Weight Gain, Food Intake, and Cachexia Indicators after Tumoural Evolution

In the gestation and lactation periods, there were no differences between DC and DL in terms of the initial and final food intake and weight gain (Table 1). The litter weight increase during the lactation period was also not different between the groups (Figure 2A), and this trend continued until adulthood. After weaning (21 days old) and at adulthood (before tumour inoculation; 120 days old), the two groups had similar body weights (Figure 2B). Therefore, the only difference between the groups was the diet administered during the gestation and lactation periods.

All groups had similar body weight, body weight gain, and food intake at the time of tumour inoculation (measurement obtained on day 120 of the experiment; Table 2). The carcass weights were not different between the control groups (C and L) on the euthanasia day (day 141; Table 2). As expected after tumour growth [34], the tumour-bearing groups had a lower body weight gain (W < C, *p* = 0.0294; WL < L, *p* < 0.0001; Table 2) and decreased food intake (W < C, *p* < 0.0001; WL < L, *p* < 0.0001; Table 2) compared with control groups. Despite having a tumour, the WL group had a higher final food intake compared with the W group (*p* = 0.0171; Table 2).

The tumour weight showed no differences between the groups (Table 2) and the gastrocnemius muscle weight decreased in the W group (*p* = 0.0089; Figure 3A), but was preserved in the WL group, in which the dams received the leucine-rich diet during the gestation and lactation periods.

The effects of tumour growth led to a decreased serum total protein (W < C and WL < L, *p* < 0.0001) and albumin (W < C, *p* = 0.0004; WL < L, *p* = 0.0002) content in the W and WL groups (Table 2). The ratio between albumin and globulin content significantly decreased only in the W group (*p* = 0.0072) and the serum triglyceride content increased in the W group (*p* = 0.0005). The WL group showed no differences in these parameters compared to the L group. The serum cholesterol decreased in both tumour-bearing groups (W < C, *p* = 0.0382; WL < L, *p* = 0.0035; Table 2).

### 3.2. Maternal Leucine-Rich Diet Attenuates the Effects of Tumour Evolution Over the Gastrocnemius Muscle Protein Synthesis and Degradation

Alkaline phosphatase activity was significantly reduced only in the W group (W < C, *p* = 0.0086) (Figure 3B), whereas in the WL group it did not change, indicating less impact on tumour growth in the muscle of rats in the tumour-bearing group with maternal leucine supplementation. Additionally, because we found differences between the loss of muscle mass in the tumour-bearing groups, protein synthesis and protein degradation were evaluated. The tumour-bearing group (W) had lower phenylalanine incorporation (W < C, *p* = 0.0048) (Figure 3C) and increased tyrosine release (W > C, *p* = 0.0104) (Figure 3D), which suggested that there was a decrease in the synthesis process and increased muscle degradation, respectively. For the tumour-bearing group with a maternal leucine-rich diet (WL), muscle protein synthesis (Figure 3C) and degradation (Figure 3D) showed no differences from the L group, and despite tumour growth, the WL group had a significantly different level of tyrosine release compared to the W group (*p* = 0.0210). These data are in accordance with the preservation of muscle weight in this group (Figure 3A).

### 3.3. Proteasome 20S Subunit Expression and Calpain and Cathepsin Activities Were Modulated in the Rats with Maternal Leucine Supplementation

The relationship between protein degradation and the proteasome subunits 20S and 19S and the 11S regulatory particle were investigated (Figure 4). The 19S, and 11S expression (Figure 4C and D) was not different among the groups. When we investigated the 20S subunit (32 and 28 kDa) (Figure 4A and B), and we found an increase in the W group compared to the C group (W > C, *p* < 0.0001 for 32 kDa; and for 28 kDa W > C, *p* < 0.0001, L > C, *p* = 0.0236, C = WL, *p* = 0.0547). Additionally, maternal leucine supplementation led to modulation of the damage effect because the 20S subunit (32 and 28 kDa) expression in the WL group was lower than that in the W group (WL < W, *p* = 0.0017 and *p* = 0.0004, respectively).

We evaluated the activity of cathepsin H, calpain, and chymotrypsin-like (Figure 5) enzymes related to muscle protein degradation. The W group had higher calpain (*p* = 0.0104; Figure 5A) and cathepsin H (*p* = 0.0306; Figure 5B) activities compared to the C group, indicating the impact on the calcium-dependent proteolysis pathway and lysosomal degradation, respectively. No differences were found between maternal leucine supplementation groups. For chymotrypsin-like activity (Figure 5C) no differences were found among the groups, in this protease related to the proteasome activity.

### 3.4. Maternal Leucine-Rich Diet Changed the mTor Muscle Protein Synthesis-Related Pathway

We investigated the mTOR pathway and the pPI3K/PI3K ratio was reduced in the W group compared to the C group (*p* = 0.0356; Figure 6A), but no differences were found between the L and WL group. The pmTOR/mTOR ratio decreased only in the W group (Figure 6B) (C vs. W, *p* = 0.0481) and the supplemented groups showed no difference. The pP70S6k/P70S6K ratio (Figure 6C), a downstream target of mTOR, showed a decrease in the W group compared to the C group (*p* = 0.0056), and the L group showed lower activation compared to the C group (*p* = 0.0410), but there was no difference between the L and WL groups (Figure 6C). Additionally, there was no difference in the p4EBP1/4EBP1 ratio (Figure 6D), another downstream protein of the mTOR pathway, among the groups.

## 4. Discussion

In this study, we investigated a preventive approach using a maternal leucine-rich diet to reduce the cancer-induced loss of muscle mass, modulating the muscle protein synthesis and degradation in adult offspring. The maternal dietary intervention during gestation and lactation periods was shown to be preserved throughout the lifetime of the offspring [4]. We evaluated the influence of the maternal diet on adulthood because the parameters evaluated for the dam and the weight increase in the offspring showed that the maternal diet did not affect these morphometric parameters. Additionally, the maternal leucine-rich diet did not affect tumour size, which is in accordance with previous studies [12,20], but it minimised the cancer cachexia-induced damage to the skeletal muscle, in this case the gastrocnemius tissue.

Because anorexia is related to weight loss and cachexia progression, an improvement in this parameter combined with reduced muscle wasting could lead to a better response to treatment [34,35]. Therefore, an important finding of this study, which likely conducted for this muscle mass improvement, was the increase in final food intake in the tumour-bearing group with a maternal leucine-rich diet (WL) compared to the tumour-bearing group (W), probably being one of the positive effects associated with this preservation of muscle tissue. Therefore, these results showed that a maternal diet supplemented with leucine could modulate the cancer cachexia-induced damage in this experimental model.

This cachexia experimental model is associated with damage processes in the rats, such as altered serum parameters and deep muscle wasting [25,36]. In this study, we verified that the maternal diet could be an alternative to minimise the cancer-induced damage in adult offspring rats, such as an improvement in triglyceride levels and the albumin-to-globulin ratio, which were improved by maternal leucine as supplementation. Additionally, decreased muscle alkaline phosphatase activity was likely related to impairment of muscle cell function, which could be linked to decreased muscle protein synthesis that is associated with increased muscle spoliation [36,37], as seen in the W group and corroborated by the decreased muscle weight. As a positive effect of maternal diet, the muscle alkaline phosphatase activity remained similar regarding protein synthesis and lower protein degradation in the WL adult offspring, which were associated with similar muscle weight, without changing the tumour weight. Therefore, the maternal leucine-rich diet could influence muscle protein turnover, as a preventive approach. 

Focusing on muscle protein signalling, the mTOR pathway is an important pathway that is related to control of protein synthesis and is impacted by cancer cachexia [8,9]. Consistent with this, cancer evolution inhibits the mTOR pathway [8,38], as shown here in the W group, and these results support decreased protein synthesis, increased protein degradation, and reduced muscle weight. However, the maternal leucine diet supplementation scheme improved this mTOR pathway, not changing the pmTOR/mTOR ratio; this reflects the maintenance of the pP70S6k/P70S6K ratio, which could be related to less muscle wasting. Therefore, the maternal leucine-rich diet modulated the mTOR signalling, which seemed to be a preventive effect.

In parallel, the skeletal muscle proteolysis is a key point to understand cachexia progression. Skeletal muscle has three main degradation pathways for protein breakdown, which are ubiquitin-proteasome, lysosomal and calcium-dependent pathways [7], and they are mainly altered in this condition. In murine models, the ubiquitin-proteasome degradation has been widely studied as the most important mechanism in muscle atrophy [8,35,39]. In this study, we found enhancement only in the 20S subunit, especially related to the proteolytic activity (20S – 32 and 28 kDa – subunit) in the W tumour-bearing group, but not on chymotrypsin-like enzyme activity, or in the proteasome subunits (19S and 11S). However, Tardif and colleagues [40] showed that the lysosomal systemcan be upregulated in patients with cachexia, which is suggested to be the promoter of the skeletal muscle proteolysis [40,41]. Similarly, calpain proteases have also been proposed to initiate the skeletal muscle degradation process, but this mechanism needs to be better investigated [42]. Confirming these findings, the Walker-256 evolution, as an experimental model of cachexia, led to muscle wasting that occurred via the lysosomal and calcium-dependent pathways (increased cathepsin H and calpains activities) only in the W group, showing the effect of the cachexia in this group. Conversely, we have shown that a maternal leucine-rich diet has preventive roles, minimising this cancer cachexia-induced injury in the WL adult offspring, which likely contributed to less muscle protein degradation. These data are in accordance with other studies that use leucine as a treatment [13,43], but here we also presented data related to prevention.

In summary, the maternally supplemented leucine group had less cachexia-induced muscle wasting compared to the control maternal diet group. This finding can be related to the muscle protein synthesis and degradation balance and by the improvement in the food intake, preserving the muscle mass. Moreover, in the recent context of cachexia as a multi-organ syndrome [1], new studies are now undergoing in our laboratory to clarify the connection between the muscle and other tissues usually impacted in this condition, such as the adipose tissue, that could also be impaired here. Additionally, contributing to our further studies, considering the maternal diet effects on the metabolic reprogramming, the role of the inflammatory and immune responses in cancer cachexia should be analysed since the muscle regeneration likely depends on the initial neutrophil and macrophage signalling as well [44]. This study had some limitations, including the length and size of the experiment. Therefore, we decided to test one previously determined leucine dose that was usually used in treatment approaches, to demonstrate here that the maternal diet with leucine can modulate cancer cachexia-induced damage, specifically in the gastrocnemius skeletal muscle.

## 5. Conclusions

In conclusion, the present study provides innovative findings about the preventive effects of a maternal leucine-rich diet on maintaining the gastrocnemius muscle mass. This maternal diet effect led to an improvement of food intake, associated with the maintenance of muscle protein synthesis and degradation by modulating the activity of proteolysis pathways and preserving mTOR pathway in skeletal muscle in adult offspring tumour-bearing rats. The results from this study echoes the concept of offspring developmental programming by maternal diet leading to further studies on the epigenetics mechanisms that culminate in these benefits.

## Figures and Tables

**Figure 1 biomolecules-09-00229-f001:**
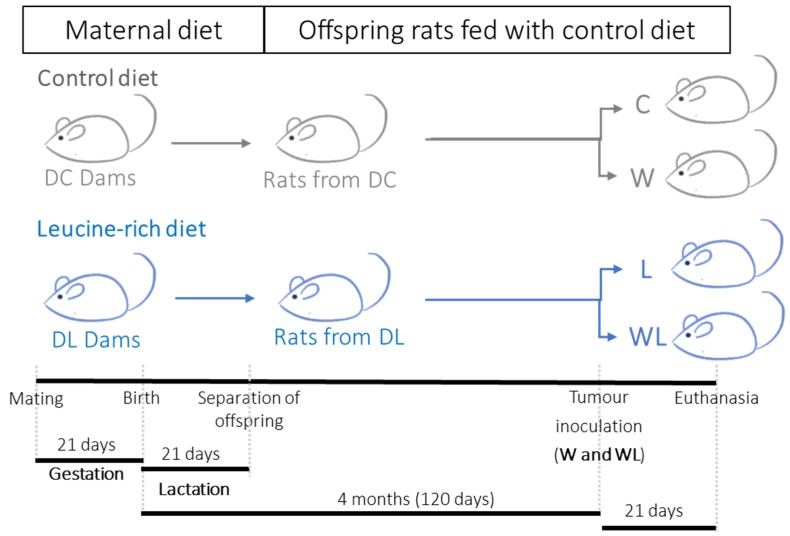
Schematic diagram of the experimental procedure. Legend: DC, dams fed control diet during gestation and lactation periods; DL, dams fed leucine-rich diet during gestation and lactation periods. Adult offspring were distributed into C, Control; W, Tumour-bearing animals; L, Group without tumour with a maternal leucine-rich diet; and WL, Tumour-bearing group with a maternal leucine-rich diet. For details, see the Materials and Methods section. DC (n = 3), DL (n = 3), and the final number of adult offspring rats (C, W, L, and WL) was 7–8 per group.

**Figure 2 biomolecules-09-00229-f002:**
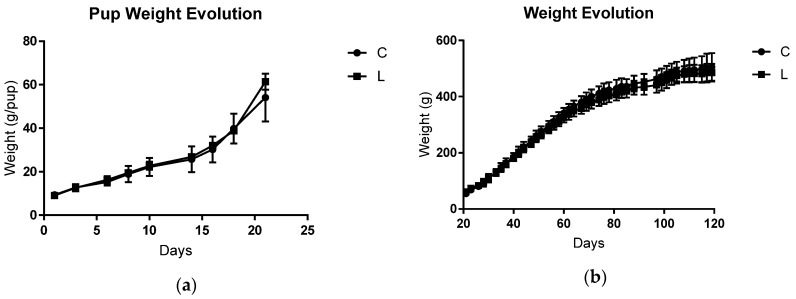
Offspring body weight evolution. (**a**) Pup weight evolution (g) during lactation until the weaning period (21 days old; litter size was adjusted to eight pups/dam). (**b**) Individual offspring weight evolution (g) after weaning until adulthood (before tumour inoculation; 21 to 120 days of life). C, offspring from a dam that was fed a control diet during its lifetime (n = 8); L, offspring from a dam fed the maternal leucine-rich diet and control diet after weaning (n = 8). The data were analysed using a *t*-test.

**Figure 3 biomolecules-09-00229-f003:**
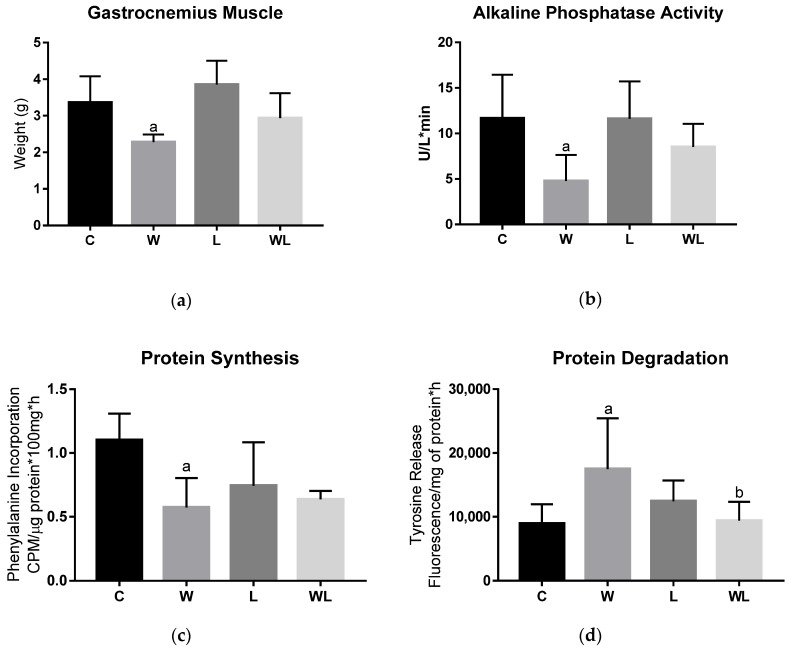
Tumour growth affected muscle protein synthesis and degradation and the phosphatase alkaline activity. (**a**) Muscle weight (g); (**b**) alkaline phosphatase activity, expressed as U/L*min; (**c**) muscle protein synthesis, expressed by the phenylalanine incorporation, CPM/µg protein*100 mg*h; (**d**) muscle protein degradation, expressed by tyrosine release, fluorescence/mg of protein*h. C, Control; W, Tumour-bearing animals; L, Group without tumour with a maternal leucine-rich diet; WL, Tumour-bearing group with a maternal leucine-rich diet. For details, see the Materials and Methods section. Values are presented as the mean ± SEM. n = 7, minimal animal per group. The data were analysed using a two-way ANOVA followed by the Bonferroni test to evaluate comparisons among groups. ^a^
*p* < 0.05, for comparison with the C group; ^b^
*p* < 0.05, for comparison with the W group.

**Figure 4 biomolecules-09-00229-f004:**
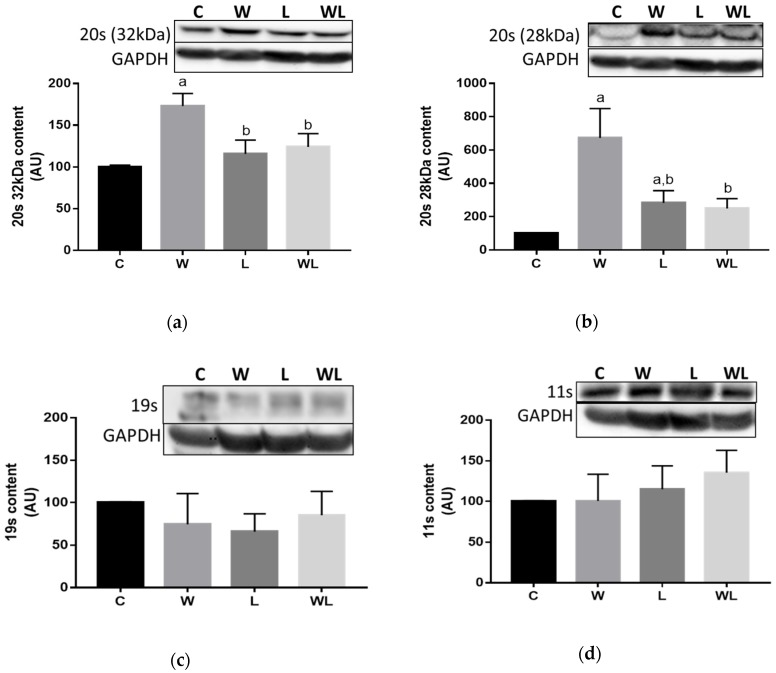
Tumour growth affected the ubiquitin-proteasome pathway. Proteasome subunits contents (**a**) 20S (32 kDa); (**b**) 20S (28 kDa); (**c**) 19S; and (**d**) 11S. The results represent bands that were quantified and normalised by GAPDH expression. Western blot images are representative of the total results obtained. Values are expressed as the relative percentage in the C group, which was set equal to 100%. C, Control; W, Tumour-bearing animals; L, Group without tumour with a maternal leucine-rich diet; WL, Tumour-bearing group with a maternal leucine-rich diet. *n* = 6, minimal animals per group. For details, see the Materials and Methods section. Values are presented as the mean ± SEM. The data were analysed using a two-way ANOVA followed by the Bonferroni test to evaluate comparisons among groups. ^a^
*p* < 0.05, for comparison with the C group; ^b^
*p* < 0.05, for comparison with the W group.

**Figure 5 biomolecules-09-00229-f005:**
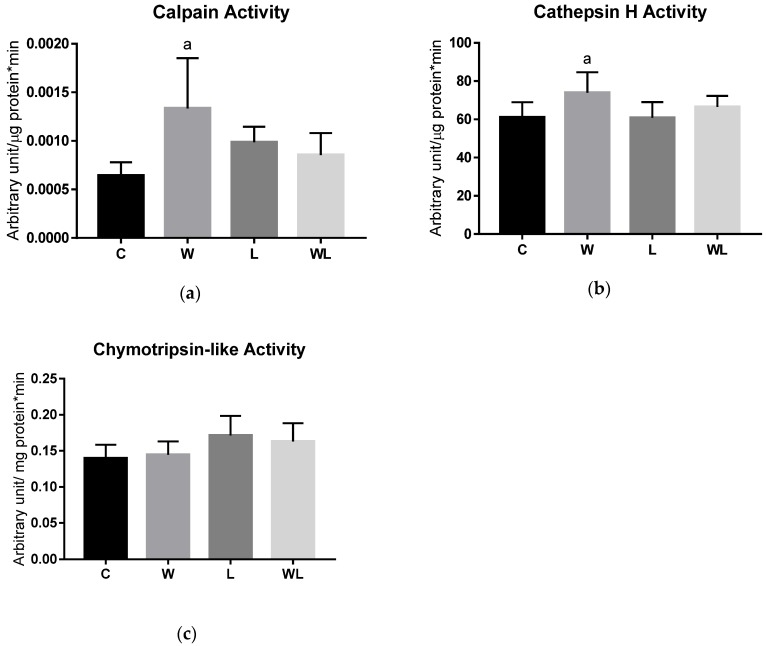
Tumour growth affected the muscle lysosomal and calcium-dependent proteolytic pathways. Enzyme activities related to the following protein degradation pathways were assessed: (**a**) calpain; (**b**) cathepsin H; and (**c**) chymotrypsin-like activities. The results are expressed as arbitrary unit/µg (or mg) protein*min. C, Control; W, Tumour-bearing animals; L, Control group with a maternal leucine-rich diet; WL, Tumour-bearing group with a maternal leucine-rich diet. For details, see the Materials and Methods section. Values are presented as the mean ± SEM. *n* = 7, minimal animals per group. The data were analysed using a two-way ANOVA followed by the Bonferroni test to evaluate comparisons among groups. ^a^
*p* < 0.05, for comparison with the C group.

**Figure 6 biomolecules-09-00229-f006:**
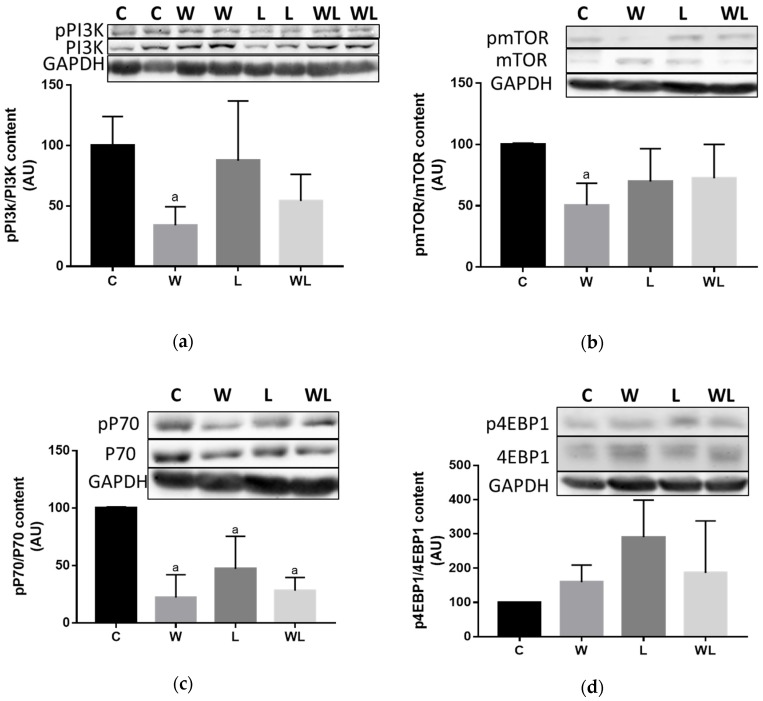
Tumour growth affected proteins related to the mTOR pathway. (**a**) pPI3K/PI3K; (**b**) pmTOR/mTOR; (**c**) pP70S6k/P70S6K; and (**d**) p4EBP1/4EBP1 ratios (percentage of arbitrary unit (AU)). The results represent bands that were quantified and normalised to GAPDH expression. Western blot images represent all analyses in the different groups. C, Control; W, Tumour-bearing animals; L, Group without tumour with a maternal leucine-rich diet; WL, Tumour-bearing group with a maternal leucine-rich diet. For details, see the Materials and Methods section. Values are presented as the mean ± SEM. *n* = 6. The data were analysed using a two-way ANOVA followed by the Bonferroni test to evaluate comparisons among groups. ^a^
*p* < 0.05, for comparison with the C group.

**Table 1 biomolecules-09-00229-t001:** Dam food intake and weight gain during the gestation and lactation periods.

	Parameters	DC	DL
Gestation	Initial Food Intake (g)	18.52 ± 3.14	18.02 ± 0.53
	Final Food Intake (g)	19.12 ± 4.14	15.12 ± 2.39
	Body Weight Gain (g)	125.30±13.35	97.33 ± 12.44
Lactation	Initial Food Intake (g)	20.43 ± 1.53	20.36 ± 3.50
	Final Food Intake (g)	28.86 ± 3.37	34.29 ± 0.75
	Body Weight Gain (g)	42.67 ± 6.69	33.33 ± 3.84

Dams’ food intake and weight gain during the gestation and lactation periods. DC, dams supplemented with control diet during the gestation and lactation period (n = 3); DL, dams supplemented with leucine-rich diet during the gestation and lactation period (n = 3). Dams’ food intake was the average measurement of the food weight offered per day (calculated as the difference of offered food minus the remaining food). Dam body weight gain was calculated using the difference between final body weight at the time of birth and initial body weight at mating. Values are presented as the mean ± SEM. The data were analysed using a *t*-test.

**Table 2 biomolecules-09-00229-t002:** Tumour growth affected body weight, food intake, and serum parameters in the tumour-bearing groups.

Morphometric and Serum Parameters	C	W	L	WL
Initial body weight (g)	514.80 ± 14.13	531.39 ± 21.66	518.06 ± 10.55	497.35 ± 10.89
Carcass weight (g)	528.41 ± 15.20	458.47 ± 21.21 ^a^	524.00 ± 12.06	414.31 ± 13.45 ^c^
Body weight gain (%)	104.56 ± 3.99	88.10 ± 1.78 ^a^	101.24 ± 1.96	83.64 ± 3.50 ^c^
Initial food intake (g)	21.48 ± 1.24	22.14 ± 1.03	23.63 ± 3.04	22.96 ± 1.23
Final food intake (g)	20.52 ± 1.27	8.46 ± 1.08 ^a^	18.08 ± 0.24	12.60 ± 0.71 ^b, c^
Tumour weight (g)	-	71.70 ± 6.94	-	80.92 ± 5.54
Total Protein (g/dL)	5.82 ± 0.28	4.32 ± 0.12 ^a^	5.14 ± 0.18	4.25 ± 0.07 ^c^
Albumin (g/dL)	3.28 ± 0.07	2.81 ± 0.06 ^a^	3.19 ± 0.07	2.80 ± 0.06 ^c^
A/G ratio	1.43 ± 0.07	1.87 ± 0.06 ^a^	1.67 ± 0.01	1.82 ± 0.09
Triglycerides (mg/dL)	118.89 ± 4.95	165.39 ± 10.20 ^a^	125.18 ± 5.41	154.50 ± 8.01
Cholesterol (mg/dL)	67.05 ± 2.65	58.91 ± 0.58 ^a^	67.69 ± 2.14	57.28 ± 1.43 ^c^

Legend: C, Control; W, Tumour-bearing animals; L, Group without tumour and with a maternal leucine-rich diet; WL, Tumour-bearing group with a maternal leucine-rich diet. Initial day corresponds to day 120 (the age of the animals before tumour inoculation); final day (Day 141) corresponds to after 21 days of tumour growth, which is the date of the euthanasia. A/G ratio corresponds to the ratio between albumin and globulin contents. For details, see the Materials and Methods section. Values are presented as the mean ± SEM (n = 7, minimal animals per group). The data were analysed using a two-way ANOVA followed by the Bonferroni test to evaluate comparisons among groups. The analysis between the two tumour-bearing groups was conducted using a *t*-test. ^a^
*p* < 0.05, compared with the C group; ^c^
*p* < 0.05, compared with the L group.

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
