# Peer review of "Maternal Leucine-Rich Diet Minimises Muscle Mass Loss in Tumour-bearing Adult Rat Offspring by Improving the Balance of Muscle Protein Synthesis and Degradation"

_biomolecules, 2019, doi:10.3390/biom9060229_

Round 1

Reviewer 1 Report

I have no further comments. The manuscript is acceptable.

Author Response

Reviewer  #1

Comment 1. I have no further comments. The manuscript is acceptable.

Answer 1. We would like to thank the reviewer #1 for the time spent analysing our manuscript and to consider it acceptable for publication.

Reviewer 2 Report

The present manuscript is focused on the prophylactic action of a maternal leucine-enriched diet on cancer-induced muscle wasting on the offspring. The concept is intriguing and relevant for human health, however, some points/data in the manuscript should be clarified in order to support the author’s conclusions.

Since the authors describe a preservation of muscle mass in WL rats as compared to W rats despite an exacerbated body weight loss, they should provide evidence of what tissue compartment is more wasted in order to preserve the muscle mass. Such point is critical in understanding the relevance for the multi-organ syndrome cancer cachexia.

The protein synthesis data are in contrast with the body and the gastrocnemius muscle data. Considering the non-tumor-bearing rats, how can the C and L rats have a similar body/muscle weight along with a different protein synthesis rate?

The numbers in table 1 are strange: how can the final food intake during gestation be so different from the initial food intake during lactation? Moreover, the numbers suggest a lower food intake during gestation and the opposite during lactation. The lack of statistical significance might be due to the low n. Can the authors give a possible explanation?

The molecular data might completely rely on the distinct feeding behavior between W and WL rats. Please discuss.

Minors:

It is unclear the reason for having parts of the text underlined in yellow.

Table 2: A/G ratio should be explained in the legend.

The GAPDH image in figure 4 is the same for both the 28 and 32 kDa 20s. Can the authors explain or provide the original full images used for the analysis?

Author Response

Reviewer  #2

Comment 1. The present manuscript is focused on the prophylactic action of a maternal leucine-enriched diet on cancer-induced muscle wasting on the offspring. The concept is intriguing and relevant for human health, however, some points/data in the manuscript should be clarified in order to support the author’s conclusions.

Answer 1: We would like to thank the reviewer #1 for the time spent analysing our manuscript. We believe that the maternal diet is a very positive approach in cancer prevention in adulthood and we thank for considering the study relevant and raising important points for the improvement of this manuscript.

Comment 2. Since the authors describe a preservation of muscle mass in WL rats as compared to W rats despite an exacerbated body weight loss, they should provide evidence of what tissue compartment is more wasted in order to preserve the muscle mass. Such point is critical in understanding the relevance for the multi-organ syndrome cancer cachexia.

Answer 2: We thanked the referee for raising this important point. It is very important focus on the cachexia as a multi-organ syndrome and, in the discussion section (highlighted in yellow) we provided a discussion of this topic showing a possible correlation between the muscle mass and adipose tissue wasting. As we have shown in our previous studies using leucine-rich diet as a treatment (the animals fed leucine-rich diet only during 21days of tumour evolution) that the muscle mass was preserved instead of the adipose tissue, but, unfortunately, we did not measure the adipose tissue waste in this study.

Comment 3. The protein synthesis data are in contrast with the body and the gastrocnemius muscle data. Considering the non-tumor-bearing rats, how can the C and L rats have a similar body/muscle weight along with a different protein synthesis rate?

Answer 3: We thanked the referee for this question, but we have shown that both non-tumour-bearing groups had similar values of initial body weight, carcass weight and muscle weight, and all the other parameters had the same profile, since we also find this similarity between the C and L group for the protein synthesis analysis (P= 0.0532), corroborating the muscle weight data.   

Comment 4. The numbers in table 1 are strange: how can the final food intake during gestation be so different from the initial food intake during lactation? Moreover, the numbers suggest a lower food intake during gestation and the opposite during lactation. The lack of statistical significance might be due to the low n. Can the authors give a possible explanation?

Answer 4: As seen in other studies [1–3], after giving birth, the food intake of the dams increases along the lactation period, in a normal condition. In our study, the gestation’s final food intake measure was in the 20th day, and the lactation’s initial food intake was measured in the 22nd day, enough time to observe that rise. Indeed, the number (n) of dams in these groups was low. However, as seen by the puppies weight evolution, in Figure 2a, the L offspring was born with a similar weight compared to the C offspring, indicating that both DC and DL dams groups were in similar conditions during this period.

Comment 5. The molecular data might completely rely on the distinct feeding behavior between W and WL rats. Please discuss.

Answer 5: We thank for raising this important point. Now we tried to improve the discussion and conclusions sections showing a connection between the difference found in the final food intake and the presented results.

Comment 6. It is unclear the reason for having parts of the text underlined in yellow.

Answer 6: The journal's policy is once the manuscript had some reviewed part this should be highlighted, and this manuscript had some yellow highlighted parts because it had included some addressed texts after the previous comments.

Comment 7. Table 2: A/G ratio should be explained in the legend.

Answer 7: We have now included the explanation in the legend of Table 2.

Comment 8. The GAPDH image in figure 4 is the same for both the 28 and 32 kDa 20s. Can the authors explain or provide the original full images used for the analysis?

Answer 8:  The GAPDH image is the same for both subunits of 20s subunits (28 and 32kDa), because the primary antibody for 20S (PW8195) probes both of them, and since they were captured in the same membrane, the GAPDH bands correspond to the same bands found to both 20s subunits.

Reviewer 3 Report

 The study is very well designed and of extremely important to the fiel of cancer cachexia. Metabolic diseases are linked to chronic inflammatory diseases, and it has become clear that the Western Diet has a critical role in the increased incidence of chronic inflammatory diseases. Importantly, many degenerative diseases, including cancer inflammation, are strongly influenced by metabolites in milk during lactation or  metabolites that can pass the placenta barrier. It is important for prevention and therapy of these diseases, to consider that epigenetic metabolic reprogramming remains present in adulthood, and might increase the susceptibility to cancer , autoimmune diseases and neurodegenerative diseases. The present study is focused on the muscle. However,  the critical role   of  metabolic reprogramming of innate immune cells that infiltrate the muscle. considered novel key drivers of cancer inflammation driven cachexia, should be discussed. Analysis of the innate inflammatory profile and function in the present study model in muscle and Bone Marrow is of great future importance to improve therapy in cancer cachexia.

Author Response

Reviewer  #3

Comment 1. The study is very well designed and of extremely important to the field of cancer cachexia. Metabolic diseases are linked to chronic inflammatory diseases, and it has become clear that the Western Diet has a critical role in the increased incidence of chronic inflammatory diseases. Importantly, many degenerative diseases, including cancer inflammation, are strongly influenced by metabolites in milk during lactation or metabolites that can pass the placenta barrier. It is important for prevention and therapy of these diseases, to consider that epigenetic metabolic reprogramming remains present in adulthood, and might increase the susceptibility to cancer, autoimmune diseases and neurodegenerative diseases.

Answer 1. We would like to thank the reviewer #3 for the time spent analysing our manuscript. We believe that the maternal diet is a very positive approach to cancer prevention in adulthood. Thus, we thank for considering the study well designed and relevant.

Comment 2. The present study is focused on the muscle. However, the critical role of metabolic reprogramming of innate immune cells that infiltrate the muscle, considered novel key drivers of cancer inflammation driven cachexia, should be discussed. Analysis of the innate inflammatory profile and function in the present study model in muscle and Bone Marrow is of great future importance to improve therapy in cancer cachexia.

Answer 2. We thank you for the suggestion and we tried to improve the discussion section, including this point as a possible future approach, and we are now thinking about our next experiments. 

Round 2

Reviewer 2 Report

The points raised in the review report are aimed at improving the manuscript. Oddly, the authors answered giving a personal opinion/response on the single points in the reply letter, with very little changes in the manuscript.

This manuscript is a resubmission of an earlier submission. The following is a list of the peer review reports and author responses from that submission.

Round 1

Reviewer 1 Report

The PI tried to prove that maternal leucine-diet supplement can decrease the severity of cachexia in offspring rat till adulthood. The idea is good but there exists severe flaws in this paper.

Does the leucine-diet change the three proposed cachexia pathway activity after delivery? and the change also lasts for 120 days?  

The body weight of the pregnant mice and offspring mice decrease significantly after leucine-diet fed, what is the reason? please clarify.

In the figure 3, author showed the protein degradation was decreased in leucine-fed mice but why their body weight was also decreased? 

In the figure 4, the western figure of 20S was not compatible between the concentration and the bar graph.

In the figure 6, the western figure of mTOS and PIK3 were also not compatible, further, the significance of pP70/P70 and was only found in comparison with control group, there is no difference among W/L/WL  groups

Reviewer 2 Report

In this manuscript, authors used the rat model with tumor transplantation to examine the effect of maternal leucine-rich diet intake during pregnancy and lactation on preventing cachexia syndrome in adult offspring. Results also indicated the regulation of mTOR signaling was associated with muscle mass loss and maternal leucine supplement. This study echoes the concept of developmental programming, in which the maternal environment make effect on organ development and stress response of offspring. Although this study provides some meaningful information to readers, there are some points needed to be clarified.

1.     Authors should provide clear hypothesis regarding how maternal leucine supplement make effects on regulating mTOR signaling in adult offspring, particularly after 120-day control diet feeding.

2.     In Table 2, compared W group, it seems that maternal leucine supplement increased the tumor weight and reduce body weight.  This result is inconsistent with findings in Fig. 3.

3.     A detailed method and illustration for ubiquitin-proteasome examination should be provided.

4.     The statistics and quantitation of Western blots should be checked. The presentation of bar charts and pictures is inconsistent.

5.     The description in page 2, line 47-49 should be checked.

6.     Authors should describe he difference between present study and authors’ recent publication (Nutrition Research 2018; 51, 29-39), a group with maternal leucine supplement was also included in that study.